# Neurotoxic Impact of Individual Anesthetic Agents on the Developing Brain

**DOI:** 10.3390/children9111779

**Published:** 2022-11-19

**Authors:** Dabin Ji, Joelle Karlik

**Affiliations:** Department of Anesthesiology, Emory University School of Medicine, Children’s Healthcare of Atlanta, Atlanta, GA 30322, USA

**Keywords:** neurotoxicity, neurodevelopment, neonatal anesthesia, pediatric anesthesia, volatile anesthetics, propofol, ketamine, benzodiazepines, thiopental, dexmedetomidine, xenon

## Abstract

Concerns about the safety of anesthetic agents in children arose after animal studies revealed disruptions in neurodevelopment after exposure to commonly used anesthetic drugs. These animal studies revealed that volatile inhalational agents, propofol, ketamine, and thiopental may have detrimental effects on neurodevelopment and cognitive function, but dexmedetomidine and xenon have been shown to have neuroprotective properties. The neurocognitive effects of benzodiazepines have not been extensively studied, so their effects on neurodevelopment are undetermined. However, experimental animal models may not truly represent the pathophysiological processes in children. Multiple landmark studies, including the MASK, PANDA, and GAS studies have provided reassurance that brief exposure to anesthesia is not associated with adverse neurocognitive outcomes in infants and children, regardless of the type of anesthetic agent used.

## 1. Introduction

Concerns about the safety of anesthetic agents in children arose in the last decade after animal studies revealed disruptions in neurodevelopment after exposure to commonly used anesthetic drugs [1]. Several subsequent human studies have demonstrated that anesthetic exposure at an early age may potentially lead to long-term cognitive and learning impairments [2,3,4]. In 2016, the U.S. Food and Drug Administration (FDA) released a Drug Safety Communication about the potential neurotoxic effects of anesthesia in children by stating that “repeated or lengthy use of general anesthetic and sedation drugs during surgeries or procedures in children younger than 3 years or in pregnant women during their third trimester may affect the development of children’s brains” [5,6]. Specifically, this warning specifically included agents that block N-methyl-D-aspartate (NMDA) receptors and/or potentiate gamma-aminobutyric acid (GABA) activity. As a result of this warning, many of the commonly used general anesthetic drugs and sedative agents were required to change their labels [6]. The FDA announced the approval of label changes in 2017, focusing on potential neurodevelopmental risk in children less than 3 years of age and for exposures over 3 h [7].

The controversy about the neurotoxic effects of anesthesia continues despite the many years of investigation. Critics have pointed out that initial FDA warnings were based on largely animal and preclinical data. Early clinical studies contained multiple limitations, such as retrospective and observational study designs, varying anesthetic protocols and exposure times, heterogenous age groups, differing outcome measures, insufficient power, and multiple sources of bias [5]. Confounders for these studies include the surgery and hospitalization itself, psychosocial interruptions such as removal from school, and personal and family stressors associated with pediatric disease.

Since the FDA warning, many landmark studies, including the General Anesthesia or Awake-regional Anesthesia in Infancy (GAS) study [8,9], the Pediatric Anesthesia NeuroDevelopment Assessment (PANDA) study [10], and the Mayo Anesthesia Safety in Kids (MASK) study [11], have provided strong evidence that brief exposure to general anesthesia at a young age does not cause profound, persistent cognitive impairments or alterations in neurodevelopment. 

Given the ongoing debate, the purpose of this review article is to assess the neurotoxic effects of individual anesthetic agents used in children. This review aims to be a brief summative review and not an extensive description of the literature. In addition, these anesthetics are very rarely given individually in clinical practice. Therefore, certain findings may not be able to be extrapolated for clinical use and must be interpreted cautiously. 

## 2. Anesthetic Agents

### 2.1. Volatile Anesthetics

Volatile anesthetics, particularly sevoflurane and isoflurane, are commonly used agents for the induction and maintenance of anesthesia in children. Volatile anesthetics have been found to act on GABA and NMDA receptors, which have been associated with adverse neurodegenerative effects on the developing brain with both cognitive and behavioral characteristics [12]. Labels on many commonly used volatile anesthetics, including isoflurane, desflurane, and sevoflurane, are now required to display the FDA warning [7]. 

Mechanisms for potential neurotoxicity from volatile anesthetics include neuronal apoptosis (1, 10–13) and synaptic changes [12,13]. Pathologic examination of the effect of volatile anesthetics in neonatal rat brains has shown varying effects on synaptic density. Early animal studies examining isoflurane and/or isoflurane/midazolam/nitrous oxide mixture showed a decreased in synaptic density [13,14]. However, separate evidence demonstrated that volatile anesthetics actually increased synaptogenesis overall [15]. A prominent difference in these studies was the age at which exposure to anesthetics occurred. These varying results imply that it is not only the agent, but the timing of exposure that impacts brain development [16]. 

The pathologic findings in these animal studies have not shown a clear resulting phenotype. Early animal studies revealed measurable negative neurocognitive effects following exposure to inhalational agents, such as sevoflurane or isoflurane, during periods of crucial brain development [1,12,17,18,19]. In a rodent study, exposure to 3% sevoflurane for 2 h a day for 3 days resulted in cognitive impairment and neuroinflammation in young mice, but not adult mice. Conversely, a two-hour exposure to 3% sevoflurane for one day did not result in cognitive impairment in either study group [20]. These animal findings suggest that a brief, single exposure may not induce detrimental neurodevelopmental effects, although repeat exposures may have negative effects. 

The first human multi-institutional, randomized controlled study to assess the neurodevelopment effects of different anesthetic techniques was the GAS study. This landmark study examined and compared children undergoing general anesthesia with sevoflurane to those receiving a regional anesthetic without undergoing general anesthesia for inguinal hernia repair [8,9]. The infants were evaluated at two years old via tasks related to problem solving, exploration, attention, concept formation, memory, and sensorimotor development. Ultimately, no difference was found in cognitive test performance at two years of age between the two cohorts [8]. At five years old, the infants were assessed with the Weschler Preschool and Primary Scale of Intelligence full-scale IQ and were found to have equivalent results [9]. The authors concluded that there was no increased risk of neurodevelopmental outcomes at two and five years of age when comparing the two cohorts [8,9]. This landmark study provides strong evidence that a single, limited exposure to general anesthesia, particularly volatile inhalational agents, during infancy does not cause profound neurotoxicity to the young brain [8,9].

The landmark PANDA study examined the longstanding effects of a single dose of inhalational general anesthetic for children less than three years old for elective inguinal hernia repair [10]. This retrospective sibling-matched cohort study compared cognitive performance between children who did and did not undergo general anesthesia early in life. The study included 100 sibling pairs which included one sibling who had been exposed to general anesthesia with either sevoflurane and/or isoflurane for a given length of time, ranging from 20 to 240 min, and the other sibling, who had not been exposed to any inhalational general anesthetic. Ultimately, investigators found no statistical difference in intelligence quotient scores between the children exposed to a volatile anesthetic and their siblings who were not exposed to volatile inhalational agents [10]. 

These landmark studies provided reassurance that a brief exposure to volatile inhalational agents does not cause profound impairments or deficits in a developing human brain. 

### 2.2. Propofol

Propofol is a commonly used anesthetic agent for general anesthesia and sedation in the pediatric population. Propofol’s mechanism of action includes decreasing the dissociation of GABA from GABA-A receptors and increasing its inhibitory effects [21]. The US Food and Drug Administration has approved the use of propofol for induction of anesthesia in children greater than three years of age, and for maintenance of anesthesia in children greater than two months of age [7,22]. However, propofol has continued to be commonly used off-label in children less than three years of age. 

Animal studies in rodents and non-human primates have raised concerns about propofol’s impact on the developing brain [23]. Similar to volatile anesthetics, propofol has varying pathologic effects on developing animal models. Propofol has also been shown to cause neuroapoptosis in the developing brain in non-human primates [24]. In these preclinical studies, fetuses had prominent neuroapoptosis in the caudal and subcortical regions of the brain. In neonates, neuroapoptosis was more prominent in the cerebral cortex [24]. In contrast, propofol has also been shown to aid in synaptogenesis in mice models [25]. These neuronal changes persisted into murine adulthood, suggesting that propofol-induced synaptic effects may cause permanent changes [16]. Preclinical studies in animal models also suggested that propofol may reduce cellular injury and increase neurogenesis [26,27]. 

Despite findings that propofol induces neurotoxicity in the developing animal brain by increasing apoptosis of neurons and of oligodendrocytes in the developing brain after exposure [24], the mechanism by which this neuronal injury occurs remain unknown. GABA agonism is believed to play a central role [16]. Other proposed mechanisms of neurotoxicity include activation of p75 neurotrophin receptor, TNF alpha, IL-6, as well impacts on dendritic spine density and other neural pathology [28]. 

Despite many varying anatomical areas and proposed mechanisms of propofol neurotoxicity, clinical findings in animal models remain limited [29]. One study examined the difference between one total dose of propofol delivered once or divided into seven exposures. Spatial learning and hippocampal function were reduced in the rats with multiple exposures, suggesting that numerous exposures may be more harmful than a cumulative dose [30]. 

While not directly studied, concern for detrimental neurodevelopment and neurocognitive effects of propofol have not been validated in a clinical study in a pediatric population. 

### 2.3. Ketamine

Ketamine is an NMDA receptor antagonist widely used for perioperative sedation and analgesia in pediatric anesthesia. Ketamine is also now used in subtherapeutic doses for the treatment of psychological disorders, including depression and schizophrenia [31]. Ketamine is included in the list of medications cited by the FDA that potentially cause deleterious neurocognitive outcomes [5].

The majority of preclinical studies on ketamine have used animal models. The mechanism by which ketamine can cause neurotoxicity and cognitive impairment is currently not completely understood. The increased susceptibility to ketamine-induced neurotoxicity in neurons in the developing brain may be explained by ketamine’s prolonged blockade of NMDA receptor channels in immature neurons when compared with mature neurons [32]. During normal brain growth-spurt periods of brain development, NMDA receptor antagonism can be triggered by neuronal apoptosis [33]. Multiple animal studies reported that ketamine-induced neuronal apoptosis is dose dependent and evolves over time [34,35,36]. However, ketamine exposure to both fetal and neonatal non-human primate brains can result in pathologic neuronal apoptosis [20]. Conversely, some animal studies even suggested that a single dose of ketamine may be neuroprotective [37]. It is likely that, like many other of the anesthetic agents, ketamine’s effect is largely based on the context, timing, and dose of its administration.

Animal studies have also examined clinical signs of ketamine neurotoxicity. Some neuronal changes were associated with clinical changes in neurocognitive testing in non-human primate models [22]. However, the majority of clinical animal studies have focused on the anti-depressive effects of ketamine, limiting their utility for the general pediatric population [31]. 

One prospective randomized study looked at the effects of a single dose of ketamine in children undergoing ventricular septal defect repair under cardiopulmonary bypass [38]. Inflammatory markers of neuronal damage (neuron-specific enolase, s100, or cytokines) were checked at certain time points after surgery following a single dose of ketamine at 2 mg/kg. At the conclusion of the study, no difference in the expression of these plasma markers of neuronal damage and inflammation was found [38]. In addition, the Bayley Scales of Infant Development-II scores between the children who received ketamine and the controlled groups revealed no significant differences [38]. Ultimately, there was no convincing evidence to suggest that ketamine was either neurotoxic or neuroprotective [38].

At this time, there is not enough clinical evidence to show an association of a single brief anesthetic exposure of ketamine in early childhood with adverse neurodevelopmental and cognitive outcomes. 

### 2.4. Benzodiazepines

Benzodiazepines are GABA agonists, which are often used for preprocedural anxiolysis, light sedation, and as part of balanced anesthesia. Medications such as midazolam, lorazepam, etc., were included in the initial FDA warning, likely due to their GABA-agonism effects [5]. However, use of benzodiazepines as a single agent is extremely rare and overall limited.

Previous animal studies have included midazolam and revealed changes in neuronal structure [14,39]. However, benzodiazepines were included along with volatile anesthetics, opioids, and nitrous oxide in these studies, and any individual effects cannot be determined. Animal research investigating the effect of only benzodiazepines is very limited.

Any detrimental neurocognitive effects of benzodiazepines have not been extensively studied, and therefore have not been validated in a clinical human setting. In a group of neonatal congenital heart disease patients, cumulative dose of benzodiazepines was associated with lower scores on the Beery-Buktenica Developmental Test of Visual Motor Integration tests at kindergarten age [40]. These medications were part of a balanced anesthetics, and while no other associations were made, the cumulative effect cannot be removed. Therefore, concern about neurodevelopmental effects of benzodiazepines have not been validated in a clinical human setting. 

### 2.5. Thiopental

Thiopental is a commonly used sedative-hypnotic barbiturate used in pediatric anesthesia. Thiopental binds to multiple ligand-gated ion channels, including GABA-A. Other involved neurotransmitters include nicotinic acetylcholine, 5-HT3, and glycine. This barbiturate is not currently listed on the FDA warning list. 

Thiopental has been shown to cause neuronal degeneration and promote apoptosis of neuronal brain cells in neonatal rats by decreasing the synthesis of GABA from glutamic acid and by increasing the expression of apoptotic proteins in neonatal rat brains [41,42]. Thiopental has also been found to have neurotoxic effects in neonatal rat brains as a result of reactive oxidative stress [43]. Interestingly, thiopental was shown to not worsen sevoflurane-induced neurotoxicity in neonatal rat models, revealing that it may not have additional detrimental neuronal effects on the developing brain when used in conjunction with sevoflurane [44]. 

At this time, detrimental neurotoxic effects from thiopental in the pediatric population have not been validated in clinical human studies. There is no clinical evidence to reveal that brief anesthetic exposure of thiopental in children leads to adverse neurodevelopmental and cognitive outcomes. 

### 2.6. Dexmedetomidine

Dexmedetomidine is a potent alpha-2 adrenoceptor agonist that has anxiolytic, sedative, analgesic, and sympatholytic properties that result in a valuable pediatric anesthetic. Dexmedetomidine is not currently listed on the FDA warning list. 

Early preclinical studies have revealed that dexmedetomidine possesses significant neuroprotective properties. Dexmedetomidine has been shown to increase anti-apoptotic proteins and reduces pro-apoptotic mediators, such as p53, which protects against ischemic neuronal and cerebral harm [45]. In addition to its anti-apoptotic effects, dexmedetomidine has been found to confer neuroprotective properties by reducing glutamate, inhibiting influx of calcium influx, and reducing activation of NMDA receptors [46]. Interestingly, there is evidence that the neuroprotective properties of dexmedetomidine include damage produced by other neurotoxic anesthetic agents. Preclinical studies have revealed conflicting results on the impact of potential dexmedetomidine-mediated neuroprotection against sevoflurane-induced neurotoxicity [47,48,49]. Co-administration of dexmedetomidine at 1 μg/kg with sevoflurane compared to administration of sevoflurane alone at similar levels of anesthesia revealed no neuroprotection and caused comparable levels of neuroapoptosis in several developing brain regions in neonatal rats [47,49]. However, one preclinical study revealed significant neuroprotection with the co-administration of dexmedetomidine at the same dose at 1 μg/kg with sevoflurane compared to administration of sevoflurane alone [48]. Interestingly, co-administration of dexmedetomidine at higher doses (5 μg/kg or higher) with sevoflurane was shown to potentiate neuroapoptosis and increase mortality in neonatal rats. Isoflurane-induced neurocognitive impairment in the thalamus and hippocampus in neonatal rats were found to dose-dependently reduce with co-administration of dexmedetomidine [50]. In this same study, dexmedetomidine protected against isoflurane-induced neurotoxicity by decreasing neuroapoptosis and inhibiting caspase-3 expression, which was promoted by isoflurane [50].

Various clinical studies have supported the notion that dexmedetomidine is a safe and effective agent for sedation in the pediatric population, especially for those children undergoing cardiac surgery, without significant associated adverse effects [51,52,53,54]. Despite the evidence of neuroprotection conferred by dexmedetomidine in preclinical studies, there are not enough robust clinical trials in in children to validate these benefits.

### 2.7. Xenon

Xenon is an odorless, colorless noble gas that is used as an anesthetic gas in the pediatric population. Xenon is currently not approved as a pediatric anesthetic agent and is not listed on the FDA warning list. Xenon acts on the NMDA receptor as an antagonist, but it does not share the similar neurotoxic and hemodynamic effects present in other NMDA antagonist anesthetic agents [55]. As an anesthetic, its use provides hemodynamic stability and has found to have neuroprotective properties, commonly by reducing neurodegeneration following neurologic insult [56,57].

Preclinical studies have revealed that xenon can reduce neurodegeneration following neuronal injury by way of upregulating synthesis of pro-survival proteins and decreasing the number of apoptotic cells [58,59]. Xenon has been found to attenuate anesthetic-induced neurotoxicity in the developing brain in in vivo studies [59]. 

Currently, there is limited evidence on the neurotoxic effects of xenon in the pediatric population. Xenon has not been approved yet as an anesthetic for children, but there is no indication that its use is associated with any detrimental neurocognitive effects in children.

## 3. Discussion

In vivo studies have revealed that neurotoxicity in the developing brain depends on multiple factors, including the number of anesthetic exposures, the type of anesthetic agent, the timing of exposure in development, and the dose of anesthetic agent [3]. Neurotoxicity was found to occur during specific periods of brain growth spurt in the developing brain [1]. These animal studies have shown that volatile inhalational agents, propofol, ketamine, and thiopental may have detrimental effects on neurodevelopment and cognitive function, but dexmedetomidine and xenon have shown to have neuroprotective properties. The neurocognitive effects of benzodiazepines have not been extensively studied, so their effects on neurodevelopment are undetermined. 

In vivo animal models may not truly portray the same pathophysiological changes and processes in children, as there are known variabilities between species. In addition, many animal models focus on pathologic findings with limited clinical correlation. At this time, data with animal models cannot entirely defend a potential association between the use of anesthetic agents and subsequent impact on neurocognitive development. Recent clinical studies describing the neurotoxic effects of anesthetic agents in children are not without their limitations, as most agents are not routinely used as a sole, primary anesthetic agent. As a result, the administration of multiple anesthetic agents, along with the existence of known confounders, has obscured efforts to clinically evaluate some of the neurocognitive side effects. A few studies, such as the GAS study, have rigorous study designs that have helped elucidate this research challenge [8,9].

It is becoming more evident that a brief, single early exposure to an anesthetic agent, regardless of the type of agent, is not associated with any profound impairments in neurodevelopmental and neurocognitive outcomes in children [60]. Most clinical studies have focused on brief and single exposure to general anesthesia. As most clinical studies are observational, appropriately accounting for confounding variables and limitations can be quite challenging. Infants who receive early, multiple, or extended exposure to anesthetic agents may have increased rates of various confounding comorbidities, including low birth weight, higher ASA status, and prematurity [61]. Inflammation, physiologic disturbances, underlying medical diagnoses, and psychologic stresses associated with surgery may also impact neurocognitive outcomes [62]. 

The MASK study further provided strong evidence that a short exposure to general anesthesia was not associated with deficits of general intelligence in later years [11]. In this matched cohort study, children who were separated into cohorts based on the number of separate exposures to general anesthesia (unexposed, single exposure, or multi-exposed). These children underwent neuropsychological testing at 8 to 12 years of age or 15 to 20 years of age [11]. General intelligence was measured by the Full-Scale intelligence quotient standard score of the Wechsler Abbreviated Scale of Intelligence. Ultimately, single exposures were not associated with deficits in neurodevelopmental or neuropsychological domains; however, multiple exposures to general anesthesia were found to correlate with decreased fine motor coordination and processing speed [11]. Similar to smaller prior studies, there was no association between exposure to general anesthesia and full-scale intelligence quotient [10,63]. 

## 4. Future Directions

Most pediatric patients undergoing anesthesia and surgery are exposed to anesthetic agents in lengths comparable to those of pediatric cohort patients in the MASK, PANDA, and GAS studies. As a result, there is reassurance that a brief, single exposure to anesthesia is not associated with adverse neurocognitive outcomes in children, regardless of the type of anesthetic agent used. However, much remains to be investigated on this topic. Studies are now trying to understand the impact of individual anesthetic agents on neurodevelopment, and how surgery-induced neuroinflammation and perioperative stress may also have an impact on brain maturation and development. The organization SmartTots, a collaborative non-profit from the International Anesthesia Research Society (IARS), FDA, and other organizations, is leading this essential research effort [64]. Future topics of research should include identifying patient risk factors for neurotoxicity, optimization of anesthetics to minimize neurotoxicity, and the contribution of physiologic factors to neurotoxicity. 

## 5. Conclusions

At this time, there is no robust clinical evidence to show that there is an association of a single brief anesthetic exposure of volatile inhalational agents, propofol, ketamine, benzodiazepines, thiopental, dexmedetomidine, or xenon in early childhood with adverse neurodevelopmental and cognitive outcomes.

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
