# Peer review of "Neurotoxic Impact of Individual Anesthetic Agents on the Developing Brain"

_children, 2022, doi:10.3390/children9111779_

Round 1

Reviewer 1 Report

I have read this article with interest.

My major concern is that the information provided in the article does not add any new or additional information in the field. The great number of review articles on the described topic exist in English scientific (pediatric as well as anesthesiology) literature and the topic has been widely discussed over last 8-5 years.. 

I am sorry to say that, but I do not think it is enough to write a short summary of the already published  comprehensive reports by the many experts in the field.

Reviewer 2 Report

While the manuscript is well written, it still needs a considerable revision to be accepted.

Major comment

How about adding the assessment of the neurotoxicity of thiopental? Thiopental is also one of the most commonly used agents in pediatric anesthesia.

Minor comment

Page 2, line 62: Please change “y-aminobutyric acid” to “gamma- aminobutyric acid” as is described in the introduction section.

Reviewer 3 Report

Timely and well written review describing neurotoxic effects of individual anesthetic agents. The article is well organized, easy to read and brief. Please see my suggestions below.

1. Page 4, line 198: Interestingly, there is evidence that the neuroprotective properties of ketamine include damage produced by other neurotoxic anesthetic agents. - did you mean to say dexmedetomidine?

2. I would consider adding a few more studies to the dexmedetomidine section where some are claiming protection against sevoflurane induced apoptosis and some studies are claiming no neuroprotection. (I'm not part of any of the below mentioned studies).

Lee JR, Joseph B, Hofacer RD, Upton B, Lee SY, Ewing L, Zhang B, Danzer SC, Loepke AW. Effect of dexmedetomidine on sevoflurane-induced neurodegeneration in neonatal rats. Br J Anaesth. 2021 May;126(5):1009-1021. doi: 10.1016/j.bja.2021.01.033. Epub 2021 Mar 12. PMID: 33722372.

Perez-Zoghbi JF, Zhu W, Neudecker V, Grafe MR, Brambrink AM. Neurotoxicity of sub-anesthetic doses of sevoflurane and dexmedetomidine co-administration in neonatal rats. Neurotoxicology. 2020 Jul;79:75-83. doi: 10.1016/j.neuro.2020.03.014. Epub 2020 May 5. PMID: 32387222.

Perez-Zoghbi JF, Zhu W, Grafe MR, Brambrink AM. Dexmedetomidine-mediated neuroprotection against sevoflurane-induced neurotoxicity extends to several brain regions in neonatal rats. Br J Anaesth. 2017 Sep 1;119(3):506-516. doi: 10.1093/bja/aex222. PMID: 28969317.

3. line 204 : various clinical studies - please provide some references or review articles about dex use in pediatric population

4. Consider adding subsections on Xenon and opioids.

Alam A, Suen KC, Hana Z, Sanders RD, Maze M, Ma D. Neuroprotection and neurotoxicity in the developing brain: an update on the effects of dexmedetomidine and xenon. Neurotoxicol Teratol. 2017 Mar-Apr;60:102-116. doi: 10.1016/j.ntt.2017.01.001. Epub 2017 Jan 6. PMID: 28065636.

Maze M, Laitio T. Neuroprotective Properties of Xenon. Mol Neurobiol. 2020 Jan;57(1):118-124. doi: 10.1007/s12035-019-01761-z. Epub 2019 Nov 22. PMID: 31758401.

Round 2

Reviewer 1 Report

Dear Authors,

Regeret to say that, but I still do not see any major changes in the article. It still does not provide any new information. It is also of little any additional value to the clinicians, which follow international recommendations with regard to safety of anesthesia in children.

Reviewer 2 Report

The manuscript has been revised well, the authors are to be commended.